# Phytosterol, Lipid and Phenolic Composition, and Biological Activities of Guava Seed Oil

**DOI:** 10.3390/molecules25112474

**Published:** 2020-05-27

**Authors:** Adchara Prommaban, Niramon Utama-ang, Anan Chaikitwattana, Chairat Uthaipibull, John B. Porter, Somdet Srichairatanakool

**Affiliations:** 1Department of Biochemistry, Faculty of Medicine, Chiang Mai University, Chiang Mai 50200, Thailand; waewaa@gmail.com; 2Department of Product Development Technology, Faculty of Agro-Industry, Chiang Mai University, Chiang Mai 50200, Thailand; niramon.u@cmu.ac.th; 3Department of Herb and Extract Business Development, Tipco Biotech Company, Prachuapkhirikhan 77000, Thailand; anan@tipco.net; 4Protein-Ligand Engineering and Molecular Biology Laboratory, National Center for Genetic Engineering and Biotechnology (BIOTEC), National Science and Technology Development Agency, Thailand Science Park, Pathum Thani 12120, Thailand; chairat@biotec.or.th; 5Department of Haematology, O’Gormond Building, Institute of Cancer Research, Huntley Street, University College London, London WC1E 6BT, UK; j.porter@ucl.ac.uk

**Keywords:** *Psidium guajava*, seed, hexane extract, lipid, phenolic compounds, phytosterols, antioxidant

## Abstract

Plant seeds have been found to contain bioactive compounds that have potential nutraceutical benefits. Guava seeds (*Psidium guajava*) are by-products in the beverage and juice industry; however, they can be utilized for a variety of commercial purposes. This study was designed to analyze the phytochemicals of the *n*-hexane extract of guava seed oil (GSO), to study its free-radical scavenging activity, and to monitor the changes in serum lipids and fatty acid profiles in rats that were fed GSO. The GSO was analyzed for phytochemicals using chromatographic methods. It was also tested for free-radical scavenging activity in hepatoma and neuroblastoma cells, and analyzed in terms of serum lipids and fatty acids. GSO was found to contain phenolic compounds (e.g., chlorogenic acid and its derivatives) and phytosterols (e.g., stimasterol, β-sitosterol and campesterol), and exerted radical-scavenging activity in cell cultures in a concentration-dependent manner. Long-term consumption of GSO did not increase cholesterol and triglyceride levels in rat serum, but it tended to decrease serum fatty acid levels in a concentration-dependent manner. This is the first study to report on the lipid, phytosterol and phenolic compositions, antioxidant activity, and the hepato- and neuro-protection of hydrogen peroxide-induced oxidative stress levels in the GSO extract.

## 1. Introduction

Guava (*Psidium guajava* L. Family Myrtaceae) is an important edible tropical fruit and a well-known herbal plant that has been widely applied in folk and traditional medicine [1,2]. The leaves are known to exhibit free-radical scavenging, inotropic, anti-glycemic, anti-hyperlipidemic, anti-hypertensive, and anti-diarrheal activities [3,4,5,6,7,8]. The pulp and peel have been known to exert anti-neoplastic effects on the induction of apoptosis and cell differentiation [9]. Guava seeds, a by-product of the beverage and juice processing industry, are abundant in dietary fiber, proteins, fats, phenolics, flavonol glycosides, glutelins, tannins, saponin and amino acids [10,11,12,13,14]. In addition, guava seed oil (GSO) obtained from red (*P. cattleianum* Sabin) and yellow (*P. cattleianum* var. lucidum Hort.) strawberry guava plants was found to contain high amounts of fatty acids, of which linoleic acid (LA) was the most abundant [15].

Different methods/agents, involving heat, boiling, roasting, detergents and organic solvents, have been used to obtain functional guava seed extracts. Additionally, GSO can be extracted by using organic solvents such as acetone, petroleum ether, ethyl acetate and *n*-hexane. For example, the sodium dodecyl sulfate extract of the seeds was shown to produce high yields of proteins that were mostly glutelins [12]. Acetone extracts of the seeds were found to contain flavonoids, phenolics and phenylethanoid glycosides [16]. The petroleum ether extract of GSO contained high amounts of linoleic acid, while the *n*-hexane extract of GSO predominantly contained linoleic acid [17,18]. Likewise, α-tocopherol and δ-tocopherol were found to be present in GSO. Notably, the quantity and nature of the tocopherols is of crucial importance regarding their oxidative stability [17]. Among organic solvents, *n*-hexane was found to be the most efficient in fractionating a wide range of lipophilic bioactive phytochemicals [19,20,21].

GSO possessed strong anti-oxidation and inhibitory activities against low-density lipoprotein peroxidation and Gram-negative bacteria [22,23]. Recently, we have revealed that the edible hexane extract of GSO is rich in linoleic acid, and contains some amounts of tocopherols, tocotrienols and phenolic compounds [24]. It is highly likely that phytochemicals in GSO present nutraceutical effects and benefits for health in humans. The aims of this study were to identify phenolic compounds, phytosterols and lipids in the GSO using very sensitive chromatographic/mass spectrometric methods, investigate the free-radical scavenging activity in hepatocytes and neuroblastoma cells, and evaluate the serum lipid levels in rats that had been fed GSO.

## 2. Results

### 2.1. Identification of Lipids

Here, we present the high-performance liquid chromatography-electrospray ionization-quadrupole-time of flight/mass spectrometry (HPLC-ESI-Q-TOF/MS) techniques, that possess high efficiency, specificity and sensitivity for the detection and characterization of tentative lipids in GSO. Though 13 peaks were resolved by chromatographic separation, only 6 peaks (No. 1–6) were identified using the Analysis Software, and these represented the tentative compounds identified as 4-hexyl-decanoic acid, sphinganine, 5S-hydroxyeicosatetraenoyl di-endoperoxide, didrovaltratum, sphingofungin B, 13,14-dihydro-19(*R*)-hydroxyprostaglandin E1, eschscholtzxanthin, tetradecan-3-one and xestoaminol C (Figure 1 and Appendix A, Table 1).

### 2.2. Identification of Phytosterols

For the detection of polar constituents, trimethysilyl (TMS) obtained from *N*-methyl-*N*-trimethylsilyltrifluoroacetamide (MSTFA) was able to derivatize certain multiple-functional groups, including hydroxyl, amine, sulfate, and carboxyl groups. Hence, the TMS derivatization of phytosterols and free fatty acids present in hexane extracts of GSO will enhance the performance of gas chromatographic/mass spectrometric (GC/MS) analysis, and provide characteristic ions in their electrospray ionization (ESI)-mass spectra. For the identification of targeted compounds, the ESI-mass fragmentations of several types of analytes were preferentially studied based on the mass spectra of the authentic standards (such as stimasterol, β-sitosterol, sitostanal and campesterol) and the internal standard cholestane. Most of the TMS derivatives and fatty acyl esters produced weak or intense molecular ions, abundant [M − 73]^+^ ions, and certain characteristic ions in their ESI-mass spectra, thereby providing easy identification. The mass fragmentation pathways of TMS-derivatized phytosterols and fatty acids were suggested and could be tentatively identified. Total ion chromatograms (TIC), and the corresponding ESI-mass spectra of typical TMS-derivatized fatty acids and phytosterols, for GSO are shown in Figure 2 and Appendix A.

Using the gas chromatography/mass spectrometry (GC/MS) scan mode, peak numbers 1–10 were identified by comparing the retention times (T_R_) of 13.25, 14.83, 14.89, 15.12, 17.70, 18.10, 20.41, 23.44, 23.70 and 24.37 min, respectively, and by comparing ESI-mass spectra against those of their authentic standards. Minor components with less than 0.1% relative abundance in the TIC were not considered of interest. As a result, they were identified as ethyl palmitate, ethyll linolenate, ethyl linoleate, ethyl stearate, linoleic acid, linolenic acid, cholestane, β-sitosterol, stigmasterol and campesterol, while sitostanal was not detected (Table 2). In stoichiometry, the amounts of β-stimasterol, β-sitosterol and campesterol were found to be 297.61, 0.22 and 11.04 mg/100 g GSO, respectively. This GC/MS method, combined with TMS derivatization, is a comprehensive chemical method for the profiling analysis and quantitation of phytosterols in GSO. Taken together, all the chromatographic analyses can provide relevant information on GSO by way of a direct comparison of its chemical composition with the biological activities in the present study. Furthermore, such information will be useful for predicting the biological and pharmacologic effects of GSO in subsequent studies.

### 2.3. Liquid Chromatographic Analysis of Phenolic Compounds

In the chromatographic profile (Figure 3), 16 small peaks were detected in the range from 100 to 700 *m*/*z*. These peaks indicated that there are at least 16 possible phenolic compounds existing in the GSO. Additional data on the high-performance liquid chromatography-single quadrupole electrospray ionization/mass spectrometry (HPLC-ESI/MS) analysis, including retention times, molecular ions and important fragment ions for tentative compounds, are presented in Appendix A. Authentic standards, including gallic acid, catechin, tannic acid, rutin, isoquercetin, hydroquinine, eriodictyol and quercetin, were analyzed and used as database, and are presented in Appendix A. In comparison with the standards, catechin, isoquercetin, eriodictyol and quercetin were detected in the GSO, while gallic acid, tannic acid, rutin and hydroquinine would not be found.

In illustration, the fragmentations of phenolic compounds in the positive ion mode, eluted at retention times of 9.47, 10.11, 12.64, 13.48, 14.02, 14.69, 16.11, 21.04, 29.53, 31.15, 32.32, 32.99, 34.47 and 35.09 min, were characterized as quinic acid, *O*-caffeoylquinic acid or chlorgenic acid, catechin, apigenin-4-*O*-glycoside, ellagic acid-*O*-methoxyglucoside, dicaffeic acid, isoquercetin, *O*-caffeoylquinic acid derivative, ellagic acid, eriodictyol, luteolin-7-*O*-rutinoside, quercetin, caffeoyl-glycosides or cinnamoyl glycosides, and di-*O*-caffeoyquinic acid, respectively (Table 3). However, two other compounds which were eluted at 17.22 and 25.84 min were not able to be identified. In term of limitation, the database library was not available for identification of the targeted compounds, and it is likely that the HPLC-ESI/MS analysis lacked sensitivity.

### 2.4. Free-Radical Scavenging Activity

It was determined that GSO treatment showed an inhibitory effect on 1,1-diphenyl-2-picrylhydrazyl radical (DPPH^•^) generation in a concentration-dependent manner (6.25–1000 mg/mL). In this determination, the inhibition by GSO was almost complete at the concentration of 200 mg/mL or 28.74 μg 6-hydroxy-2,5,7,8-tetramethylchroman-2-carboxylic acid (Trolox) equivalent (TE)/mL. In comparison, Trolox and α-tocopherol were found to be even more effective in scavenging DPPH^•^ than GSO, and in this determination the inhibition levels were dependent upon the concentrations of 20–60 µg/mL, and were complete above 60 µg/mL (Figure 4). In addition, GSO was measured at a half maximal effective concentration (EC_50_) against Trolox, and the results revealed that GSO decreased the initial DPPH^•^ concentration by 50% at a concentration value of 139 g GSO/g [DPPH^•^].

Furthermore, GSO and α-tocopherol were found to reduce reactive oxygen species (ROS) levels in human hepatocellular carcinoma (HepG2) cells, in a concentration-dependent manner (*p* < 0.05 at 100 and 200 µg/mL), when compared with non-treated cells. In this determination, GSO was found to be less effective at equal concentrations of α-tocopherol (Figure 5, left). Similarly, both GSO and α-tocopherol dose-dependently suppressed the elevation of ROS levels in hydrogen peroxide (H_2_O_2_)-induced human neuroblastoma (SH-SY5Y) cells, for which the degree of inhibition was significant at 200 μg/mL of α-tocopherol (Figure 5, right). It is likely that antioxidant compounds, including polar phenolic compounds (such as quinic acid and its derivatives, chlorogenic acid and its derivatives, and hydrolysable tannins), phytosterols (such as stimasterol and campesterol) and lipids (such as linoleic acid), exist as a consequence of GSO-protected oxidative stress in HepG2 and SH-SY5Y cells, consequently preventing neurodegenerative diseases.

### 2.5. Bioavailability of Serum Lipids in GSO-Fed Rats

As shown in Table 4, serum levels of total cholesterol did not change significantly in the rats that had been fed DI (control), CO (reference oil) (30 g linoleic acid equivalent (LAE)/kg) and GSO (6 and 30 g LAE)/kg) for 90 d, irrespective of gender. The serum cholesterol levels were neither influenced by the GSO and CO feeding nor GSO doses when compared with DI. Throughout the study, serum triglyceride levels were found to be higher in male rats than in female rats (*p* > 0.05) among all gender-based rat groups, while they were not different in mixed gender groups and tended to decrease over the course of the study. For 90 d, serum triglyceride levels were decreased to a greater degree in the CO (30 g LAE/kg) treatment (Δ 28.5 mg/dL) and the GSO (30 g LAE/kg) treatment (Δ 16.5 mg/dL) than in the DI group (Δ 7.6 mg/dL) (*P* < 0.05), while serum triglyceride levels were found to have increased in the GSO (6 g LAE/kg) treatment (Δ 24.3 mg/dL).

Here, we used a very sensitive high-performance liquid chromatography/fluorescence detection (HPLC/FLD) method, together with alcoholic acid hydrolysis, for the quantitation of derivatized fatty acids. Authentic fatty acids, including α-linolenic acid (ALA), arachidonic acid (AA), palmitoleic acid (PLA), linoleic acid (LA), stearic acid (SA) and oleic acid (OA) (100 µM each), were used to calibrate the column, on which they were positioned separately at T_R_ of 19.90, 24.71, 26.72, 30.24, 46.46 and 49.66 min, respectively (Appendix A). Using the specific T_R_ of the standard fatty acids, serum concentrations of PLA, OA, SA, LA, ALA and AA were calculated, and are shown in Table 5. The findings presented in Table 5 show very low or undetectable concentrations of serum PLA and ALA in all rat groups. Feeding LA-rich CO and GSO (30 g LAE/kg) for 90 d did not increase serum levels of LA, but did show a tendency to decrease them when compared to the control DI. Importantly, there was a tendency of CO (30 g LAE/kg) and GSO (30 g LAE/kg) feeding to significantly decrease the serum levels of OA, SA and AA in all rat groups (mixed genders), for which the GSO was more efficient than the CO when compared with the DI control group. In addition, GSO (30 g LAE/kg) was found to have lowered the serum SA level to a greater degree than GSO (6 g LAE/kg) (*p* < 0.05). These results imply that GSO consumption potentially lowers the plasma triglyceride levels, and can modulate the serum levels of fatty acids (e.g., stearic, oleic and arachidonic acids), similarly to CO consumption at an equal concentration.

## 3. Discussion

Natural products derived from several parts, of plants including leaves, roots, bark, rhizome, stock, pulp and seeds, have provided unparalleled sources of chemical diversity that possess bioactive molecules of valuable interest. Polyunsaturated fatty acids (PUFA) are rich in tree-born seed oils, and have been claimed to be rich in lipophilic antioxidants that are highly susceptible to oxidation, possibly leading to the generation of rancid oil and secondary lipid peroxides. So far, hyphenated analytical methods, such as GC/MS and HPLC/MS, have been applied for the efficient detection and characterization of targeted molecules. Likewise, GC/MS is a sensitive method for the comprehensive characterization of volatile small molecules such as fatty acids. Predominantly, HPLC-ESI/MS and high performance liquid chromatography-electrospray ionization/mass spectrometry/mass spectrometry (HPLC-ESI/MS/MS)) are powerful analytical techniques, with high sensitivity and accuracy, that can be used to determine the compound profile of plant materials and natural products [33]. Moreover, HPLC-ESI-Q-TOF/MS provides higher resolution, faster speeds and less solvent consumption, leading to a rapid and sensitive characterization of certain unexpected natural products [34,35].

Our recent findings have shown that the hexane extract of GSO was abundant with linoleic acid (most abundant at 69.95% of total fatty acids), followed by oleic acid, palmitic acid, stearic acid, arachidic acid and α-linolenic acid. Consistently, GSO (*Psidium gaujava* L.) extraction with *n*-hexane solvent revealed a high content of linoleic acid (60.03% of total fatty acids) as the main fatty acid component [18]. In comparison, petroleum ether extract of GSO gave a higher yield of linoleic acid (78.4% of total fatty acids) [17], suggesting that both the method employed and the solvent administered can affect the different percentage yields of linoleic acid. In studies on lipids, reverse-phase HPLC-ESI/MS is a robust and popular technique that is commonly used. In the literature, xestoaminol C obtained from a Fiji sponge *Xestospongia* sp. was reported to be an extremely active agent against parasites and microbes [36]. Not surprisingly, eschscholtzxanthin, which is the predominant pigment carotenoid in poppy petals, was found in the hexane extracts of GSO [37]. In this study, we have detected the presence of certain possible lipids, including 4-hexyl-decanoic acid, sphinganine, 5S-hydroxyeicosatetraenoyl di-endoperoxide, didrovaltratum, sphingofungin B, 13,14-dihydro-19(*R*)-hydroxyprostaglandin E1, eschscholtzxanthin, tetradecan-3-one and xestoaminol C, in GSO. Surprisingly, none of these compounds have ever been reported to be isolated in the hexane extracts of GSO, but some compounds have been described. For instance, 4-hexyl-decanoic acid (isopalmitic acid) is a natural major branch-chain saturated fatty acid that is present in the leaves of *Abies pindrow*, and probably produced as a lipophilic adsorbent during hexane extraction [38,39]. Interestingly, sphingoganin and sphingofungin, existing in natural seed oils, and xestoaminol C found in New Zealand Brown Alga *Xiphophora chondrophylla*, showed strong anti-candidiasis and anti-tubercular activities [40,41,42]. Notably, Eschscholtzxanthin is one of the red pigment carotenoids that is synthesized in the leaves of several plants as a response to photoinhibitory conditions during winter acclimation and displays antioxidant activity [43]. Consistently, we have demonstrated the anti-leukemic and anti-plasmodium activities of GSO [24]. Recently, we reported that the hexane extract of GSO was abundant with α-tocopherol (23.0 mg/kg) and β-tocotrienol (70.5 mg/kg) [17,44]. With regard to potential health benefits, phytosterols could play an important role in facilitating certain biological activities such as free-radical scavenging and plasma lipid-modulating effects. Here, we have analyzed GSO and found the presence of certain phytosterols, such as β-sitosterol (297.61 mg/100 g), stigmasterol (0.22 mg/100 g), campesterol (11.04 mg/100 g), and other neutral lipids. In comparison, hexane extracts of *Panax quinquefolium* ginseng and *Cajanus cajan* seed oils contained phytosterols such as squalene, oxidosqualene, campesterol, stigmasterol, clerosterol, β-sitosterol, β-amyrin, δ(5)-avenasterol, δ[5,24(25)]-stigmasterol, lupeol, δ(7)-sitosterol, δ(7)-avenasterol, 24-methylenecycloartanol and citrostadienol [45,46]. Additionally, hexane extracts of *Alyssum homolocarpum* seed oil were abundant with β-sitosterol (3.3 mg/g) and campesterol (0.86 mg/g), which readily transverse the blood-brain barrier [47]. Moreover, a variety of phytosterols, and their contents in hexane-extractable seed oils derived from Cajanus cajan, nutmeg, white mustard, anise, coriander and caraway, have been demonstrated [46,48].

We have revealed that the hexane extracts of GSO contained a total phenolic content of 45.57 ± 0.97 µg gallic acid equivalent (GAE)/g of GSO [43]. In this research, the HPLC-ESI single quadrupole/MS combined with correlation analysis of measured versus predicted mass spectra was used to afford the rapid characterization of small organic molecules, particularly the phenolic compounds present in GSO. Possible compounds included quinic acid, chlorogenic acid, catechin, apigenin-4-*O*-glycoside, ellagic acid-*O*-methoxyglucoside, dicaffeic acid, isoquercetin, chlorogenic acid derivative, ellagic acid, eriodictyol, luteolin-7-*O*-rutinoside, quercetin, caffeoyl-glycosides or cinnamoyl-glycosides and di-*O*-caffeoyquinic acid. In the case presented here, some compounds were identified after being compared with corresponding compounds that had been previously identified in natural products. Nonetheless, ESI single quadrupole/MS has traditionally presented certain limitations, such as unavailable authentic compounds as references, an incomplete database library, less ability to observe compounds of low polarity, and low sensitivity and mass resolution for detection when compared with HPLC-ESI/MS/MS or HPLC-ESI-QTOF/MS. In addition, the raw mass spectrometry data showed that many fragments were difficult to isolate for interpretation. Evidently, caffeoylquinic acid (chlorogenic acid) isomers, quinic acid derivatives, di-*O*-caffeoylquinic acid and caffeoylquinic acid derivatives were found in different proportions in the extracts obtained from *Prunus domestica*, *Salicornia gaudichaudiana, Galphimia glauca, Gymnaster koraiensis, Artemisia princeps* Pampanini and *Foeniculum vulgare* Mill, depending on organic solvent extractions [49,50,51,52,53,54,55]. Additionally, apigenin, luteolin and their glycosides are acknowledged as the predominant phenolic compounds in the hexane extracts of plant seeds, and these exert strong antioxidant or free-radical scavenging abilities, or altogether effects [46,56,57,58]. It is likely that the chlorogenic acid, caffeic acid derivatives, apigenin and its glycosides, exhibit antioxidant, free-radical scavenging and anti-inflammatory effects, suggesting that GSO would be significantly beneficial for living cells and human health.

In fact, PUFA, tocopherols, tocotrienols, phytosterols and phenols are the most important natural antioxidants that are present in vegetable and seed oils. Hence, the hexane extracts of seed oils that possess anti-oxidative compounds can also exert free-radical scavenging properties. Among available antioxidant assays, the DPPH^•^ scavenging assay is a simple colorimetric method that can be applied for the study in both hydrophobic (e.g., hexane extract) and hydrophilic environments. In this study, we found that GSO was able to decrease levels of DPPH^•^ in aqueous environments, and H_2_O_2_-induced oxidative stress in HepG2 and SH-SY5Y cells in a concentration-dependent manner, even though it was less efficient than the reference antioxidants Trolox (water-soluble vitamin E analogue) and α-tocopherol, respectively. Obviously, GSO presented EC_50_ of the DPPH^•^ scavenging activity at a dose of 139 g oil/g [DPPH^•^] or 19.97 μg TE/g [DPPH^•^], which then exerted antioxidant activity. Similarly, the researchers have shown that GSO reduced the initial DPPH^•^ concentration by 50% (EC_50_) at a dose of 12.9 g oil/g [DPPH^•^] [17], which is potentially more powerful than our experiments with GSO. However, our GSO experiments demonstrated a lower value of EC_50_ than other oils extracted from walnuts, peanuts, almonds, hazelnuts and pistachio nuts [59]. Moreover, we have proposed that the hepato- and neuro-protective effects acknowledged in GSO would be mediated by active phytosterols and phenolic compounds, via scavenging radicals that are generated during the production of oxidative stress. Notably, antioxidant activity may occur as a result of the presence of hydrophobic compounds (e.g., carotenoids, tocopherols, tocotrienols and linoleic acid per se) [60,61,62], along with the polar phenolics [63] that are present in certain plant seed oils including GSO, as was observed in this study. Zhang and Liu have demonstrated that the extracts derived from millets (*Setaria italic* and *Panicum miliaceum*) contain chlorogenic acid, caffeic acid, xanthophylls and zeaxanthin which could scavenge peroxyl radicals in 2,2-azobis-amidinopropane-induced HepG2 cells [64,65]. Likewise, the chlorogenic acid and di-*O*-caffeoylquinic acid present in plant extracts showed protective effects against oxidative stress-induced hepatic (HepG2), neuroblastoma (SH SY5Y), and nepheochromocytoma (P-12) cell damage [53,54,66,67]. Moreover, β-sitosterol and its glucoside obtained from the *Gastrocotyle hispida* extract exhibited DPPH^•^-scavenging activity in a concentration-dependent manner [68]. Pretreatment with phytosterols, including β-sitosterol, daucosterol and pectolinarin obtained from *Cirsium setidens* and *Aster scaber* extracts, considerably decreased ROS levels in H_2_O_2_-induced neuroblastoma (SK-N-SH) cells [69].

Although GSO have exhibited important nutraceutical properties, such as high contents of linoleic acid, tocopherols and tocotrienols, anti-oxidation and wound healing effects, long-term consumption of the GSO may lead to dyslipidemia and increased levels of atherogenic index. Evidently, GSO (30 g LAE/kg) and CO (6 g LAE/kg) treatments were found to lower the serum levels of triglycerides significantly, when compared to treatments without GSO. Similarly, the GSO and CO treatments could significantly decrease the serum levels of fatty acids, including oleic, stearic and arachidonic acids, when compared to treatments without GSO. In these treatments, the GSO (30 g LAE/kg) seemed to display a level of efficiency that was equal to the CO at an equal concentration but was greater than that of the GSO (6 g LAE/kg). From previous instances, long-term consumption of LA-rich corn oil and ALA-rich perilla oil revealed a trend of reducing serum triglyceride and cholesterol levels in rodents, when compared with the DI control. However, the consumption of flaxseed oil and CO did not change the serum cholesterol and triglyceride levels in humans. In the study, serum ALA was undetectable in all rat groups, while serum LA levels were lower in CO and GSO groups than in the DI group. A previous study has demonstrated that grape seed oil and corn oil, that were rich in linoleic acid or omega-6 fatty acids, tended to decrease serum cholesterol levels in rats [70]. Similarly, linoleic acid-rich niger seed oil was found to display a hypolipidemic effect through the facilitation of lipid transportation and metabolism, possibly lowering the risk of cardiovascular disease development [71]. Consistently, a combination diet of proteins, perilla seed oil, linoleic acid and α-linolenic acid (per oral) was found to decrease plasma triglyceride and cholesterol levels in rats [72]. Importantly, the supplementation of caffeoylquinic acid (or chlorogenic acid), which is rich in the extracts obtained from chicory (*Cichorium intybus*) seeds and GSO, decreased levels of the triglyceride and atherogenic index, but increased levels of antioxidant capacity in the serum of rats that were fed with a high fructose/glucose diet [73]. Conversely, feeding rats with a berry seed oil-supplemented diet did not significantly influence the plasma levels of triglycerides, while the total cholesterol values, which included high-density and low-density lipoprotein fractions, were not changed [74]. The results suggest that GSO rich in α-linoleic acid, phytosterols and chlorogenic acid could be utilized as a source of functional lipids, in the synthesis of essentially longer n-6 fatty acids, and the protection of oxidative stress-induced hepatocytes and neuronal cells.

## 4. Materials and Methods

### 4.1. Chemicals and Reagents

Arachidonic acid (AA), boron trifluoride (BF_3_), 4-bromomethyl-7-methoxycoumarin (Br-MMC), 18-crown-6, 2′,7′-dichlorodihydrofluorescein diacetate (DCFH-DA), 1,1-diphenyl-2-picrylhydrazyl (DPPH), Folin-Ciocalteu reagent, formic acid, hydrogen peroxide, linoleic acid (LA), α-linolenic acid (ALA), oleic acid (OA), palmitic acid (PA), palmitoleic acid (PLA), phosphoric acid, stearic acid (SA), α-tocopherol, and 6-hydroxy-2,5,7,8-tetramethylchroman-2-carboxylic acid (Trolox) were purchased from Sigma-Aldrich Company (St. Louis, MO, USA). Acetonitrile, acetic acid, ethanol, ethyl acetate, *n*-hexane, heptadecanoic acid, *n*-heptane, isopropanol, methanol, *n*-pentane (the highest pure HPLC grade), *N*-methyl-*N*-trimethylsilyltrifluoroacetamide (MSTFA, LiChropur^™^, purity ≥ 98.5%) and deionized water (DI) (Milli-DI^®^, Resistivity at 25 °C > 1 MΩ∙cm) were purchased from Merck Chemical Company (Merck KGaA, Darmstadt, Germany). Dulbecco’s modified Eagle medium (DMEM), Ham’s F12 Nutrient, fetal bovine serum (FBS), penicillin-streptomycin (10,000 U/L) and trypsin-ethylene diamine tetra acetic acid (trypsin-EDTA) were purchased from Gibco Technologies (Thermo Fisher Scientific, Waltham, MA, United States). Corn oil (CO) (density 0.91–0.93 g/mL, compositions of 2–3 g SA, 11–13 mg PA, 25–31 g OA, 59 g LA, 1 g ALA, 0.3 g *trans*-fats and 14.3 mg α-tocopherol in 100 g of oil) was purchased from a supermarket that was located in the city of Chiang Mai.

### 4.2. Preparation of GSO Extract

Dried guava seeds (*P. guajava* var Pan See Tong) (200 g) were ground and extracted with *n*-hexane (1000 mL) in a Soxhlet apparatus (boiling point in the range of 60–80 °C) for 8 h. The extract was then filtered through a Buchner funnel with Whatman’s No. 1 filter paper and concentrated at 60 °C using a rotatory evaporator [75]. The GSO obtained was decolorized with activated charcoal and stored in a plastic container in the dark at 4 °C for further study. From the chromatographic analyses, we found that the GSO contained linoleic acid, palmitic acid and oleic acid (69.95%, 6.14% and 10.47% of total fatty acids, respectively), with a n3/n6 ratio of 1:224; α-tocopherol (23.0 mg/kg); β-tocopherol (1.5 mg/kg); γ-tocopherol (1.4 mg/kg); β-tocotrienol (70.5 mg/kg); δ-tocotrienol (17.4 mg/kg) and γ-tocotrienol (4.0 mg/kg) [24].

### 4.3. High-Performance Liquid Chromatography-Electrospray Ionization-Quadrupole Time-of-Flight/Mass Spectrometry for Lipids

GSO was analyzed for its polar phenolic compounds at the Central Laboratory, Faculty of Agriculture, Chiang Mai University, Chiang Mai, Thailand, using the HPLC-ESI-QTOF/MS method that was established by Gu and colleagues with slight modifications [35]. The HPLC instrument was equipped with an ESI-QTOF/MS machine (Agilent Acquisition SW version 6200 series TOF/6545 series LC/Q-TOF), QTOF Firmware Version 25.698 (Agilent Technologies, Santa Clara, CA, USA). Mobile phase A (acetonitrile) and mobile phase B (0.1% formic acid) were degassed at 25 °C for 15 min. The GSO (20 mg) was constituted in 1.0 mL of the A:B mixture (1:1, *v*/*v*), filtered using a syringe filer (polyvinylidene fluoride type, 0.45 μm pore size, Millipore, MA, USA) and put into HPLC vials. The flow rate was set to be 0.35 mL/min, the injection volume was measured at 10 μL for each sample, and the running time was 60 min. Chromatographic separation was carried out on a column (InfinityLab Poroshell 120 EC-C18 type, 2.1 mm × 100 mm, 2.7 μm, Agilent Technologies, Santa Clara, CA, USA) that was regulated thermally at 40 °C. The ESI-MS/MS spectra were recorded using the Agilent Q-TOF mass spectrometer (Agilent Technologies, Santa Clara, CA, USA). In the MS system, nitrogen gas nebulization was set at 45 pounds per inch^2^ with a flow rate of 5 L/min at 300 °C, and the sheath gas was set at 11 L/min at 250 °C. In addition, the capillary and nozzle voltage values were set at 3.5 kV and 500 V, respectively. A complete mass scan was conducted as a mass to charge ratio (*m*/*z*) ranging from 200 to 3200. All the operations, acquisition and analysis of the data were monitored using Agilent LC-Q-TOF-MS MassHunter Acquisition Software Version B.04.00 (Agilent Technologies, Santa Clara, CA, USA) “Find by Be” algorithm to generate a list of accurate mass matches-compounds. Peak identification was performed in positive modes using the library database, and the identification scores were further selected for characterization and *m*/*z* verification.

### 4.4. Trimethylsilylation Derivatization-Gas Chromatography/Mass Spectrometry of Phytosterols

#### 4.4.1. Derivatization

The trimethylsilyl (TMS) derivatization-GC/MS of phytosterols for GSO was performed at the Central Laboratory (North Branch), Department of Land Development, Ministry of Agriculture and Cooperation, Chiang Mai, Thailand using the method established by Lee et al. with slight modifications [76]. Briefly, GSO (100 μL) was evaporated to dryness under the nitrogen stream. The dried residue was re-dissolved in 1% pyridine in ethyl acetate (40 μL) and MSTFA (100 μL), and derivatized at 80 °C for 30 min. After cooling, the resulting solution was diluted with 0.4 mL of ethyl acetate and 60 μL of cholestane (internal standard). A 4-μL aliquot of the resulting solution was directly injected into the GC/MS system. All of the derivatives were analyzed using GC-MS scan mode.

#### 4.4.2. Gas Chromatography/Mass Spectrometry

GC-MS analysis was performed in full scan mode on the GC system (Agilent Technologies Model 6890N, Deutschland, GmbH, Waldbronn, Germany) that was directly coupled to the MS detector (Agilent Technologies Model 5973 inert, Palo Alto, CA, USA). Chromatographic separation was performed using a capillary column (DB-5MS, a dimension of 30 m × 0.25 mm, 0.25 μm film thickness, an Agilent J&W Scientific (Folsom, CA, USA). Ultra-high purity helium was used as the carrier gas at a flow rate of 1.5 mL/min, with an inlet temperature 270 °C and an auxiliary temperature of 280 °C, for a running time of 35 min. The sample solution was injected in split mode (split ratio 10:1) at 280 °C. The electrospray ionization (ESI) energy was set at 70 eV. For the single quadrupole MS system, the temperatures of the ion source and the interface were set at 150 °C and 230 °C, respectively. The mass scan ranged from 40 to 500 *m*/*z*. The selected ion monitoring (SIM) mode was set at 272, 382, 394 and 486 *m*/*z* for TMS-derivatized sterols, while the scanned mode was set in a range of 40 to 500 *m*/*z* for TMS-derivatized unknown values. The oven temperature was programmed as follows: 80 °C (held for 3 min), ramped to 110 °C at 10 °C/min (held for 5 min), increased to 190 °C (held for 3 min), ramped to 220 °C at 10 °C/min (held for 4 min), and increased to 280 °C at 15 °C/min (held for 13 min). In the phytosterol groups, β-stigmasterol, sitosterol, sitostanol and campesterol were identified using authentic reference standards. In addition, the mass fragments of the analytes were compared with the data of known compounds using a comprehensive mass-spectral library (Wiley version 7.0, www.wiley.com) for identification of the targeted molecules. In terms of method validation, the limit of detection (LOD) was 0.5 mg/kg (ppm), the limit of quantitation (LOQ) was 1.20 mg/kg (ppm), and the recovery value was 70–110%.

### 4.5. High-Performance Liquid Chromatography-Electrospray Ionization/Mass Spectrometry of Phenolic Compounds

Qualitative analysis of phenolic compounds was carried out at the Central Laboratory (North Branch), Department of Land Development, Ministry of Agriculture and Cooperation, Chiang Mai using the HPLC/MS method established by Cuyckens and colleagues with slight modifications [77]. The HPLC system (Agilent Technologies 1100 Series, Deutschland GmbH, Waldbronn, Germany) consisted of a quaternary pump (G1311A), an online vacuum degasser (G1322A), an autosampler (G1313A), a thermostated column compartment (G1316A) and a photodiode array (PDA) detector (G1315A). The outlet of the PDA was coupled directly to the atmospheric pressure ESI interface of the mass spectrometer (MS) detector (Agilent Technologies 1100 LC/MSD SL, Palo Alto, CA, USA) through a flow splitter (1:1). In further analysis, GSO (20 mg) was constituted in 1.0 mL of the mixture of solvent A (acetonitrile) and solvent B (10 mM formate buffer pH 4.0) (1:1, *v*/*v*) and filtered through a syringe filter (polytetrafluoroethylene membrane, 25 mm diameter, 0.45-μm pore size, Corning^®^) before being used, and was then injected (20 μL) into the system. Chromatographic separation was carried out on a column (LiChroCART RP-18e, 150 mm × 4.6 mm, 5 μm particle size; Purospher STAR, Merck, Darmstadt, Germany) operated at 40 °C. The mobile phases A and B were run at a flow rate of 1.0 mL/min under the gradient program of 100% B (0% A) for an initial period of 5 min, 0–20% A from 5 to 10 min, 20% A from 10 to 20 min, 20–40% A from 20 to 60 min, 40% A for 3 min, and followed by an initial 100% B for 5 min. PDA detection was set at 270 nm. MS analysis was done in positive ESI mode, and spectra were acquired within the mass to charge ratio (*m*/*z*) ranging from 100 to 700. For the single quadrupole MS system, the ESI energy was set at 70 eV, while the temperatures of the ion source and the interface were set at 150 °C and 230 °C, respectively. Nitrogen was used as the nebulizing, drying and collision gas. The capillary temperature was set to 320 °C, the nebulizer pressure was set to 60 psi, and the drying gas flow rate was set to 13 L/min. Capillary voltages were set to 3500 V (positive) and 150 V (negative). The oven temperature was programmed as follows: 80 °C (held for 3 min), ramped to 110 °C at 10 °C/min (held for 5 min), increased to 190 °C (held for 3 min), ramped to 220 °C at 10 °C/min (held for 4 min), and increased to 280 °C at 15 °C/min (held for 13 min). Accurate mass measurements were performed by the auto mass calibration method using an external mass calibration solution (ESI-L Low Concentration Tuning Mix; Agilent calibration solution B). Herein, the LOD, LOQ and recovery value were found to be 0.5 mg/kg, 1.20 mg/kg and 70–110%, respectively. The chromatographic and mass spectrometric analyses and prediction of the chemical formula, including the exact mass calculation, were performed by Mass Hunter software version B.04.00 build 4.0.479.0 (Agilent Technology, Santa Clara, CA, USA). Available authentic phenolics (1 mg/mL each) such as gallic acid, catechin, tannic acid, rutin, isoquercetin, hydroquinine, eriodictyol and quercetin were also analyzed and used as database. In addition, MS data were searched for in published literature repositories.

### 4.6. Determination of Free-Radical Scavenging Activity

Antioxidant activity of GSO was assayed using the DPPH^•^-scavenging method [78,79]. Briefly, GSO (0–1000 mg/mL ethyl acetate) or Trolox (0–250 μg/mL ethanol) was mixed in equal volumes with 0.2 mM DPPH^•^ solution and incubated at 25 °C in the dark for 30 min. The OD of the colored product was measured at 515 nm against the reagent blank. The percentage of free-radical scavenging activity was calculated by applying the following formula:% DPPH^•^ scavenging = [1 − (OD_sample_ − OD_blank sample_)/OD_control_] × 100(1)

The values of GSO and Trolox decreased the initial DPPH^•^ concentration by 50% (EC_50_) and were calculated by graphically plotting the percentage of the remaining DPPH^•^ concentrations [17].

In addition, cellular ROS scavenging activity was determined in human hepatocellular carcinoma (HepG2) and neuroblastoma (SH-SY5Y) cells by using the fluorescent dichlorofluorescein (DCF)-fluorometric method [80]. In principle, DCFH-DA substrate simply diffuses into the cells and is hydrolyzed by cellular esterase to produce 2′,7′-dichlorofluorescein (reduced), which will be subsequently oxidized by existing ROS to a green fluorescence DCF product. Fluorescence intensity (FI) is directly proportional to the amount of ROS in the cells.

In assay, HepG2 cells were cultured in DMEM supplemented with 10% FBS, penicillin G (100 U/mL) and streptomycin (100 µg/mL) at 37 °C in a humidified atmospheric 5% CO_2_ incubator. The cells (10^4^ cells/well) were incubated with GSO or α-tocopherol at concentrations of 0–200 µg/mL, which were previously diluted with DMSO (control) for 24 h at 37 °C. Afterwards, the treated cells were incubated with a 10 µM DCFH-DA solution at 37 °C for 30 min and washed twice with PBS. After being challenged with 100 µM hydrogen peroxide (H_2_O_2_), kinetic FI was measured at wavelengths of λ_ex_ 485 nm and λ_em_ 530 nm using a flow cytometer (Guava^®^ easyCyte, Merck, Darmstadt, Germany).

SH-SY5Y cell line was purchased from the American Type Culture Collection (ATCC^®^ CRL2266™, Manassa, VA, USA). Cells were cultured in DMEM and Ham’s F12 Nutrient Mixture (ratio 1:1, by volume) supplemented with 20% (*v*/*v*) FBS, 100 U/mL penicillin and 100 µg/mL streptomycin, and maintained at 37 °C in a 5% CO_2_ incubator [81]. Similarly, SH-SY5Y cells (1 × 10^4^ cells/well and viability > 80%) were seeded in 96-well plates for 24 h, then treated with GSO or α-tocopherol (12.5–200 μg/mL) or DMSO for 24 h at 37 °C. Afterwards, the treated cells were incubated with 10 µM DCFH-DA at 37 °C for 30 min, washed twice with PBS solution and challenged with 100 μM H_2_O_2_. The FI values were then measured at wavelengths of λ_ex_ 485 nm and λ_em_ 530 nm with 1-h kinetic mode using a flow cytometer (Guava^®^ easyCyte, Merck, Darmstadt, Germany).

### 4.7. Analysis Serum Lipids in Guava Seed Oil-Fed Rats

The protocol for the study involving animals was approved by the Ethical Committee for Animal Studies of the Medical Faculty, Chiang Mai University (Protocol Number 43/2558). Wistar rats were randomly divided into 4 groups (5 male and 5 female rats in each group), and fed with a nutritionally-balanced commercial diet (No. C.P. 082, Perfect Companion Group Co. Ltd., Bangkok, Thailand) comprised of crude protein 24%, fat 4.5%, fiber 5%, minerals (Ca 1.0%, Na 0.20%, K 1.17%, Mg 0.23%, Mn 171 ppm, Cu 22 ppm, Zn 100 ppm, Fe 180 ppm, Se 0.1 ppm) and vitamins (A 20,000 IU/kg, D 4000 IU/kg, E 100 mg/kg, B_1_ 5 mg/kg, B_2_ 20 mg/kg, B_6_ 20 mg/kg, B_12_ 0.036 mg/kg, niacin 100 mg/kg, folic acid 6 mg/kg, biotin 0.4 mg/kg, pantothenic acid 60 mg/kg). The rats were orally administered with DI, CO [30 g linoleic acid equivalent (LAE)/kg], and GSO (6 g LAE and 30 g LAE/kg) for 90 d. Fasting blood was collected on days 0 and 90, and serum was separated for the analysis of total cholesterol and triglyceride levels using a Randox^®^ automated analyzer (Randox Laboratories Ltd., County Antrim, United Kingdom) according to the manufacturer’s instructions.

In addition, serum fatty acid levels were quantified using the high-performance liquid chromatography/fluorescence detection (HPLC/FLD) method [82]. With regard to the assay, serum (100 µL) was firstly spiked with an internal standard of 5 mM heptadecanoic acid (20 µL), and then incubated with Dole’s reagent (isopropanol: *n*-heptane: 2 M phosphoric acid = 40:10:1 by volume) (500 µL) at room temperature for 5–10 min. Secondly, the mixture was incubated with *n*-heptane (200 µL) and water (300 µL), and centrifuged at 1000× *g* for 5 min. Thirdly, the upper-layer heptane extract (200 µL) was aspirated, evaporated using nitrogen gas, and incubated with a derivatizing reagent (200 µL) made of 10 mg Br-MMC, 26.5 mg 18-crown-6 and 100 mg potassium carbonate in 10 mL of acetonitrile at 60 °C for 15 min, in order to produce a fluorogenic methyl-7-methoxycoumarin fatty acid (MMC-FA) derivative. The derivative solution was passed through a 0.45 μm nylon-membrane filter and analyzed using the HPLC/FLD system. The conditions included a column (C18 type, 4.6 mm × 250 mm, 5 μm particle size, Agilent Technologies, Santa Clara, California, United States), a mobile-phase solvent (acetonitrile:DI = 85:15, *v*/*v*), a flow rate of 1.5 mL/min, fluorescence detection (λ_ex_ 325 nm, λ_em_ 398 nm) and a data recorder/integrator using Millenium 32 HPLC Software (version 3.2, Waters Corporation, Milford, MA, USA, 2000). Serum fatty acid values were identified by comparison with the specific T_R_ of the authentic fatty acids including α-linolenic, arachidonic, palmitoleic, stearic and oleic acids. Concentrations of serum fatty acids were determined from the standard curves constructed from different concentrations.

### 4.8. Statistical Analysis

Data were analyzed using the SPSS program (IBM SPSS Statistics V 22.0, (IBM Coporation, Ormonk, NY, USA, 2013 shared license by Chiang Mai University) and are presented as mean ± standard deviation (SD) or mean ± standard error of mean (SEM) values. Statistical significance was determined using one-way analysis of variance (post-hoc = Tukey-HSD). *p* < 0.05 was considered significant.

## 5. Conclusions

In conclusion, our studies show that guava seed oil (*Psidium guajava*) contains phenolic compounds and phytosterols, which are free-radical scavenging and have protective effects against cell damage in hepatic- and neuro-cell lines. Chronic oral administration to rats at high doses was associated with significant lowering of plasma triglycerides, without alteration in serum cholesterol, similar to that seen with corn oil administration at the same dose. The potential clinical relevance of the findings merits further investigation.

## Figures and Tables

**Figure 1 molecules-25-02474-f001:**
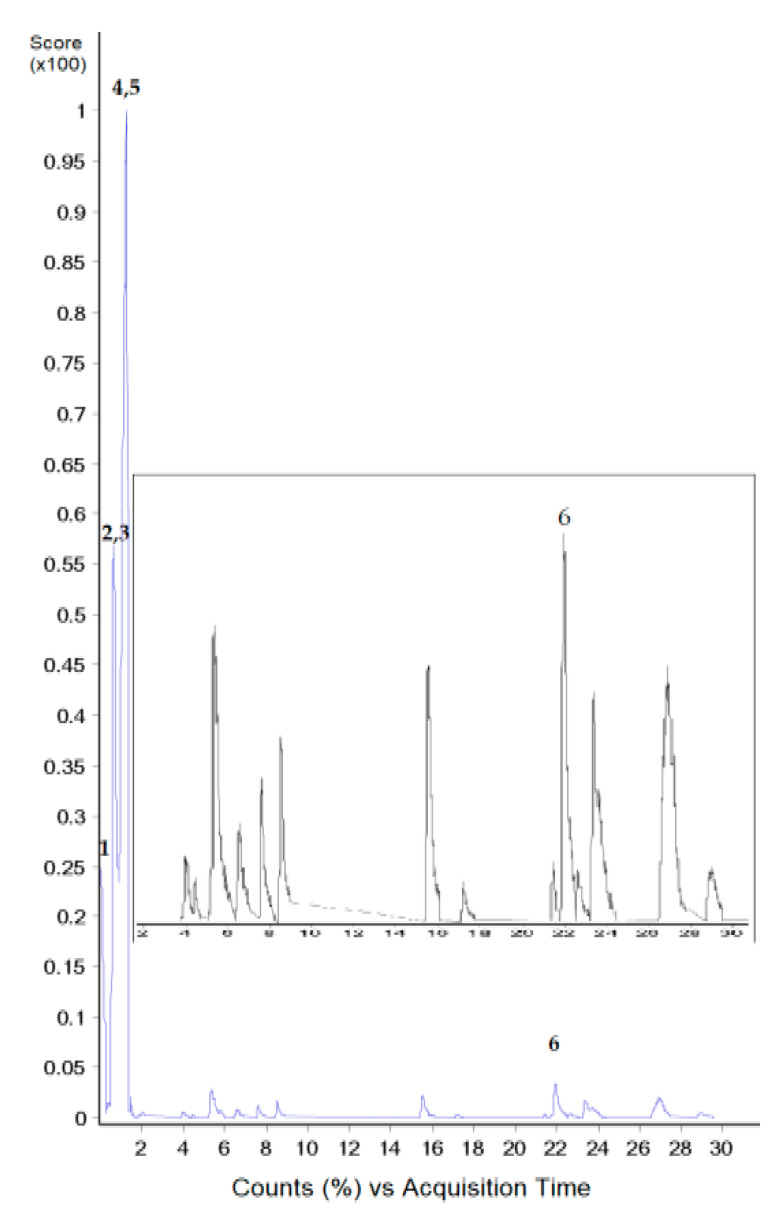
HPLC-ESI-QTOF/MS profile of lipids in guava seed oil. Inserted figure shows a magnification of area between 2 and 30 min.

**Figure 2 molecules-25-02474-f002:**
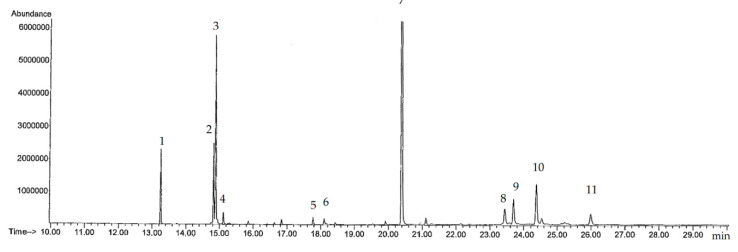
Total ion counts of phytosterols and lipids in guava seed oil analyzed using trimethylsillyl (TMS) derivatization-gas chromatography/mass spectrometry (GC/MS).

**Figure 3 molecules-25-02474-f003:**
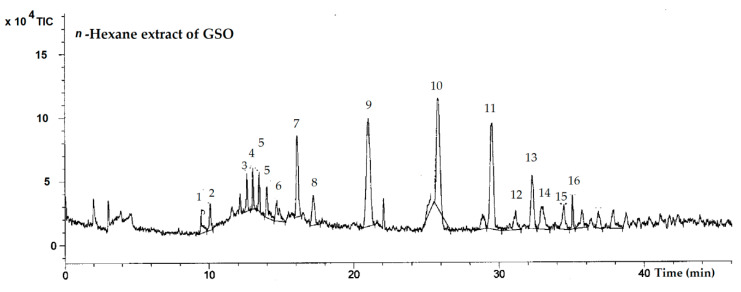
Total ion counts of HPLC-ESI/MS analysis for GSO. Peak identities are numbered in Table 3.

**Figure 4 molecules-25-02474-f004:**
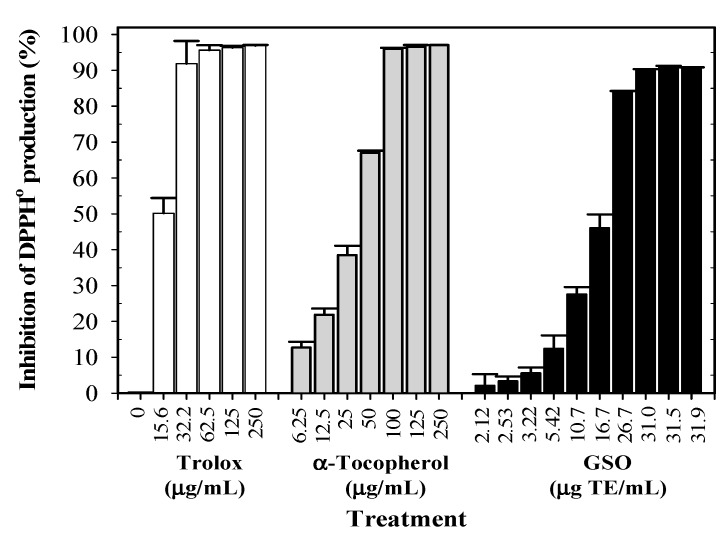
Inhibition of 1,1-diphenyl-2-picrylhydrazyl radical (DPPH^•^) generation by guava seed oil, α-tocopherol and Trolox. Data obtained from two independent experiments performed in triplicate are expressed as mean ± standard deviation (SD).

**Figure 5 molecules-25-02474-f005:**
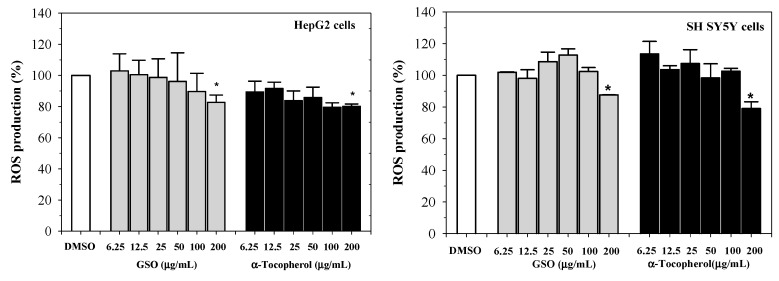
Free-radical scavenging activity of reactive oxygen species in hydrogen peroxide-induced human hepatocellular carcinoma (HepG2) and neuroblastoma (SH-SY5Y) cells by guava seed oil and α-tocopherol. Data obtained from two independent experiments performed in triplicate are expressed as mean ± SD values. * *p* < 0.05 when compared to non-treated cells. Abbreviations: GSO = guava seed oil, ROS = reactive oxygen species.

**Table 1 molecules-25-02474-t001:** Qualitative analysis for lipids in guava seed oil using HPLC-ESI-QTOF/MS.

Peak	T_R_	Target	Mass Error	Molecular Formula	Exact Mass	Observed Mass (*m*/*z*)			Identification
No.	(min)	Score	(ppm)		(g/mol)	[M + H]^+^	[M + NH_4_]^+^	[M + Na]^+^	
1	0.576	96.14	3.86	C_16_H_32_O_2_	256.24	257.24	274.27	279.22	4-Hexyl-decanoic acid
		96.14	3.54	C_16_H_35_NO_2_	273.27	274.27	291.30	296.25	Sphinganine
2	1.051	82.26	1.82	C_20_H_34_O_8_	402.23	403.23	420.26	425.21	5S-Hydroxyeicosatetraenoyl di-endoperoxide
		89.23	−3.36	C_22_H_32_O_8_	424.21	425.21	442.24	447.20	Didrovaltratum
3	1.066	95.18	1.45	C_20_ H_39_NO_6_	389.28	390.28	407.31	412.27	Sphingofungin B
4	1.094	96.85	2.19	C_20_ H_36_O_6_	372.25	373.25	390.28	399.25	13,14-Dihydro-19(*R*)-hydroxyprostaglandin E1
5	1.177	52.09	−1.18	C_40_H_54_O_2_	566.41	567.41	585.44	591.42	Eschscholtzxanthin
6	21.93	98.82	2.03	C_14_H_28_O	212.21	213.22	230.24	235.20	Tetradecan-3-one
		98.82	1.88	C_14_H_31_NO	229.24	230.24	247.27	252.22	Xestoaminol C

Abbreviations: *m*/*z* = mass to charge ratio, ppm = part per million, T_R_ = retention time.

**Table 2 molecules-25-02474-t002:** Identification of phytosterols and lipids in guava seed oil using TMS derivatization-GC/MS.

Peak No.	T_R_ (min)	TIC	Exact Mass (g/mol)	Molecular Formula	Observed Mass (*m*/*z*)	Error (%)	Identification
1	13.25	2277755	284.5	C_18_H_36_O_2_	284	−0.18	Ethyl palmitate
2	14.83	2423828	308.5	C_20_H_36_O_2_	308	−0.16	Ethyll linolenate
3	14.89	5411461	310.5	C_20_H_38_O_2_	310	−0.16	Ethyl linoleate
4	15.12	359612	312.5	C_20_H_40_O_2_	312	−0.16	Ethyl stearate
5	17.77	3836542	352.6	C_18_H_32_O_2_	353	0.11	Linoleic acid TMS
6	18.10	17168	350.6	C_18_H_30_O_2_	352	0.4	Linolenic acid TMS
7	20.41	18788155	372.7	C_27_H_48_	372	−0.19	Cholestane
8	23.44	452445	486.9	C_29_H_50_O	484	−0.6	β-Sitosterol TMS
9	23.70	770310	485.8	C_29_H_48_O	485	−0.16	Stigmasterol TMS
10	24.37	12299199	472.9	C_28_H_48_O	472	−0.19	Campesterol TMS
11	25.99	324939	-	-	-	-	Unknown

Abbreviations: *m*/*z* = mass to charge ration, TIC = total ion count, TMS = trimethylsilyl, T_R_ = retention time.

**Table 3 molecules-25-02474-t003:** HPLC-ESI/MS identification of phenolic compounds for guava seed oil.

Peak No.	T_R_ (min)	TIC	Exact Mass (g/mol)	Molecular Formula	Observed Mass (*m*/*z*)	Error (%)	Possible Constituents/Compounds	References
1	9.47	153000	192.1	C_7_H_12_O_6_	[M − H]^+^ 194.1	1.03	Quinic acid	[25]
2	10.11	174000	354.3	C_16_H_18_O_9_	[M − H]^+^ 354.9	0.17	*O*-Caffeoylquinic acid	[25,26]
3	12.64	185000	290.3	C_15_H_14_O_6_	[M − H]^+^ 298.9	2.88	Catechin	Authentic standard
4	13.48	152000	432.1	C_21_H_20_O_10_	[M − H]^+^ 433.3	0.23	Apigenin-4-*O*-glycoside	[27,28]
5	14.02	254000	496.2	C_21_H_21_O_14_	[M − H]^+^ 497.2	0.2	Ellagic acid-*O*-methoxyglucoside	[29,30]
6	14.69	254000	342.3	C_18_H_14_O_7_	[M − H]^+^ 342.9	0.15	Dicaffeic acid	[26,28]
7	16.11	649000	464.1	C_15_H_10_O_7_	[M − H]^+^ 454.3	2.15	isoquercetin	Authentic standard
8	17.22	318000	Unknown	Unknown	ND	ND	Unknown	-
9	21.04	160000	452.1	Unknown	[M − H]^+^ 453.2	0.27	*O*-Caffeoylquinic acid derivative	[25]
10	25.84	129000	Unknown	Unknown	[M − H]^+^ ND	ND	Unknown	-
11	29.53	138000	302.2	C_14_H_6_O_8_	[M − H]^+^ 302.5	0.1	Ellagic acid	[31,32]
12	31.15	306000	288.25	C_15_H_12_O_6_	[M − H]^+^ 297.0	2.95	Eriodictyol	Authentic standard
13	32.32	543000	594.5	C_27_H_30_O_15_	[M − H]^+^ 595.5	0.17	Luteolin-7-*O*-rutinoside	[24]
14	32.99	379000	302.2	C_15_H_10_O_7_	[M − K]^+^ 340.3	0.26	Quercetin	Authentic standard
15	34.47	261000	382.0	C_17_H_18_O_10_	[M − H]^+^ 383.3	0.34	Caffeoyl-glycosides or cinnamoyl glycosides	[26]
16	35.09	180000	516.45	C₂₅H₂₄O₁₂	[M − H]^+^ 517.3	0.16	di-*O*-Caffeoyquinic acid	[26]

**Table 4 molecules-25-02474-t004:** Serum levels of total cholesterol and triglyceride from rats (5 male and 5 female each) treated with deionized water, corn oil (30 g LAE/kg) and guava seed oil (6 g and 30 g LAE/kg) for 90 days. Data are expressed as individual and mean ± SD values. **^&^**
*p* < 0.05 when compared with the level at the baseline (day 0).

Serum Lipids	Time	DI			CO (30 g LAE/kg)			GSO (6 g LAE/kg)			GSO (30 g LAE/kg)		
		5 M	5 F	5 M, 5 F	5 M	5 F	5 M, 5 F	5 M	5 F	5 M, 5 F	5 M	5 F	5 M, 5 F
**Total cholesterol (mg/dL)**	Day 0	64.4 ± 8.0	61.8 ± 11.5	63.1 ± 9.4	65.6 ± 10.7	66.2 ± 6.9	65.9 ± 8.5	67.4 ± 11.2	72.6 ± 11.6	70.0 ± 11.1	62.4 ± 4.2	69.2 ± 12.7	65.8 ± 9.6
	Day 90	66.8 ± 13.5	51.8 ± 8.5	59.3 ± 13.3	62.2 ± 4.8	65.4 ± 6.1	63.8 ± 5.5	72.8 ± 16.4	62.2 ± 4.8	67.5 ± 12.7	74.0 ± 6.5	62.6 ± 10.1	68.3 ± 10.0
**Triglyceride (mg/dL)**	Day 0	87.0 ± 24.2	33.2 ± 8.2	60.1 ± 33.1	100.4 ± 24.6	44.8 ± 4.3	72.6 ± 33.7	70.0 ± 17.4	41.8 ± 10.0	55.9 ± 20.0	89.4 ± 34.2	60.8 ± 9.9	75.1 ± 28.1
	Day 90	66.2 ± 19.4	38.8 ± 11.7	52.5 ± 20.9	39.8 ± 6.4	48.4 ± 7.9	44.1 ± 8.1 ^&^	118.6 ± 45.1	39.8 ± 6.4	79.2 ± 51.4	77.4 ± 20.2	39.8 ± 9.8	58.6 ± 24.8 ^&^

Abbreviations: CO = corn oil, DI = deionized water, GSO = guava seed oil, LAE = linoleic acid equivalent, TC = total cholesterol, TG = triglyceride.

**Table 5 molecules-25-02474-t005:** Levels of fatty acids of serum obtained from rats treated with deionized water, corn oil and guava seed oil for 90 days. Data are expressed as mean ± SD values. * *p* < 0.05 when compared with deionized water; **^#^**
*p* < 0.05 when compared with the levels at a lower concentration.

Fatty Acid Levels	DI			CO (30 g LAE/kg)			GSO (6 g LAE/kg)			GSO (30 g LAE/kg)		
	5 M	5 F	5 M, 5 F	5 M	5 F	5 M, 5 F	5 M	5 F	5 M, 5 F	5 M	5 F	5 M, 5 F
Palmitoleic acid (mg/dL)	0.72 ± 0.29	ND	0.72 ± 0.29	ND	ND	ND	ND	ND	ND	ND	ND	ND
Oleic acid (mg/dL)	1.79 ± 0.85	1.00 ± 0.35	1.39 ± 0.74	0.57 ± 0.19	0.75 ± 0.19	0.66 ± 0.20 *	1.04 ± 0.37	0.61 ± 0.16	0.82 ± 0.35 *	0.54 ± 0.17	0.62 ± 0.07	0.58 ± 0.13 *
Stearic acid (mg/dL)	4.39 ± 1.38	2.55 ± 0.77	3.47 ± 1.44	1.57 ± 0.65	1.84 ± 0.37	1.71 ± 0.52 *	2.74 ± 0.43	1.38 ± 0.25	2.06 ± 0.79 *	1.37 ± 0.22	1.33 ± 0.15	1.35 ± 0.18 *^,#^
Linoleic acid (mg/dL)	1.70 ± 0.66	0.90 ± 0.31	1.30 ± 0.64	0.97 ± 0.41	1.41 ± 0.37	1.19 ± 0.43	1.05 ± 0.35	0.59 ± 0.15	0.82 ± 0.35	0.82 ± 0.24	0.78 ± 0.11	0.80 ± 0.18
α-Linolenic acid (mg/dL)	ND	ND	ND	ND	ND	ND	ND	ND	ND	ND	ND	ND
Arachidonic acid (mg/dL)	0.74 ± 0.20	0.55 ± 0.13	0.64 ± 0.19	0.34 ± 0.13	0.40 ± 0.06	0.37 ± 0.10 *	0.48 ± 0.10	0.40 ± 0.04	0.44 ± 0.08 *	0.35 ± 0.05	0.43 ± 0.13	0.39 ± 0.10 *

Abbreviations: CO = corn oil, DI = deionized water, F = female, GSO = guava seed oil, M = male, ND = not detectable.

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
