# Peer review of "Phytosterol, Lipid and Phenolic Composition, and Biological Activities of Guava Seed Oil"

_molecules, 2020, doi:10.3390/molecules25112474_

Round 1
Reviewer 1 Report
In my opinion, the authors still have not satisfactorily answered to my comments.
- If the results on triglyceride levels are not significant, writing “lipid modulation” is still not supported by the data. “No significant difference” is not the same thing as “modulation”.
- Figure 1 is not clear. All analyte peaks are eluted within 22 min, and the chromatogram line ends at 30 min; however, the run time is set at 60 min, and this makes the chromatogram almost unreadable.
- The fact that other authors have previously used a water-based assay on lipid matrices does not make this approach more valid. The authors should demonstrate experimentally that using a water-based assay produces reliable results on a lipid-based matrix. Or better, they should simply use a lipid-based assay.
In my opinion, the manuscript still requires major revision.
Reviewer 2 Report
In manuscript is lot formal mistakes, the authors must improved. In material and methods line 357 must be state also. Line 360 1L. Line 479 is new chapter? Figure 6 must be in different place not after conclusion. It is graphical abstract? In results chapter 2.1 results are compared with references, I think that this part must be in discussion or results and discussion can be together. The figure 1 is in very poor quality.
Round 2
Reviewer 1 Report
The authors have satisfactorily answered to my comments. The manuscript can be published in its current form.
Author Response
Please see the attachment

This manuscript is a resubmission of an earlier submission. The following is a list of the peer review reports and author responses from that submission.
Round 1
Reviewer 1 Report
The article is very interesting from a biological point of view, in particular with the changing of serum lipid profiles of rats and less interesting from phytochemical point of view.
Phytochemical composition of the same extract was already pubblished by the authors in the article: Phytother Res. 2019 Oct;33(10):2749-2764. Also DPPH activity was already pubblished in different articles. I suggest mentioning these articles and change the setting of the phytochemical part.
I have some doubts that the extract obtained with hexane is in the form of oil, please check and, eventually, change delete the term oil in the manuscript.
Change in the title and short title "compositions" with "composition".
Change in the abstract "plants seeds" with "plant seeds".
Reviewer 2 Report
Although the manuscript style is far from ideal, the introduction is not adequate, there are several serious drawbacks in the results, and the discussion is very poor, it would be probably acceptable after major revision. However, majority of the results is not novel. The authors forgot to mention that the data on fatty acid profile, total phenolics content as well as tocopherol content and antioxidant activity have already been published in various journals. Most importantly, the identical data on fatty acid profile and total phenolic content of hexane extracted guava seed oil presented in this manuscript have already been published by the same authors in 2019 in Phytotherapy Research 33 (10): 2749-2764, doi: 10.1002/ptr.6449. I consider this as a fraud and I can not recommend the manuscript for publication in any journal. Please find below incomplete list of my comments.
Introduction
Line 43:
The correct abbreviation of the authority is “L.”, not “Linn.”
Line 50:
The authority should be (if) stated at the first appearance of a plant scientific name. The correct authority is “Sabine“ in this case. The species name should be spelled “cattleyanum”. Please see the article of Tuler et al. (2018, TAXON 67 (6): 1194–1198) for details.
Calling the psidium fruits “strawberries” can be misleading. I recommend keeping the term “strawberry guava”.
Line 56:
Please pay attention to the grammatical structure of the sentence.
Line 59:
What “few studies”? These should be cited here together with the main findings. As it is, the purpose of this study is not adequately justified here. A bit more information on the traditional use, bioactivity and composition of guava seeds/seed oil would be appreciated.
Results
Line 70: The percentage content of the most abundant FA would be more appropriate than the retention times.
Abbreviations should be defined at the first appearance in the text.
Line 71: It is more than probable that any plant oil is varied in terms of FA type and content.
Line 72: Here, the total percentage of unsaturated FA would be more appreciated instead of LA content (reported in Table 1), eventually n-3/n-6 ratio.
Table 1:
The “LA” abbreviation is not defined in the legend. However, since the Molecules applies a single-column typesetting, there is no need to abbreviate the fatty acid names. The table will occupy the same space in the manuscript.
The sum of total fatty acids identified should be included.
Figure 1:
I recommend reformatting the chromatogram in order to reduce its height and to improve the visibility of the peaks. Focus on the time range with identified peaks only. The retention times should not cover the peaks.
What were the percentage contents of the remaining peaks? What about the peak at 17.327 min, the two peaks between 29 and 31 min, the bunch of peaks between 45 and 48 min, and the group at the end of the chromatogram? Why these peaks were not identified? (I don’t mention the “invisible” part full of overlapping retention times, I consider it just integrated baseline) Some of the peaks seem to be higher than ALA and AA peaks. Please clarify.
Line 90:
Were the compounds identified or “possibly identified”? Please clarify.
The comment on the phenolics results is unsatisfactory. The Figure 2 is showing over 20 peaks whereas the result report is limited to very brief statement on two compounds identified, most probably with very low probability. The HPLC analysis drawbacks/complications/difficulties are ignored also in the discussion.
Were any authentic standards used? When the identification was not successful is there at least a possibility to exclude some phenolics, to declare them as not present in the GSO?
Line 94: Free-radical scavenging activity
Is there any justification for testing concentrations as high as 100 %? (1000 mg/mL)
The results should be recalculated to trolox equivalents so that the results are comparable with literature.
Line 104:
GSO was less effective than DMSO? Please clarify. If DMSO was used it should be mentioned in the methodology.
Figure 5.
The figure is just a mess, the data are absolutely unreadable.
What is M0 and M3?
Discussion
The discussion part is very weak, without proper critical evaluation, citing only general findings and statements from other studies whereas a comparison with real concrete data is missing.
Line 180
There are numerous other oils exerting DPPH radical scavenging activity. However, the results (values) must be compared. The data on the activity and composition of guava seed oil or of seed oils from related species are highly recommended in this regard together with a well-known example of a species with high activity.
Line 186
I do not think carotenoids, tocopherols or tocotrienols can be classified as hydrophobic lipids.
Methods
Lines 254 to 262:
The HPLC analysis is included in the subchapter 4.5 Quantification of phenolic compounds. However, no quantification was carried out using HPLC.
More information on the identification method should be provided.
Line 277: Analysis plasma lipids in GSO-fed rats
What was the diet of the rats? What were the sampling times? Why a positive control was not included?
Reviewer 3 Report
This manuscript deals with the assessment of the composition and biological activity of a guava seed oil extract.
The paper is interesting and mostly well-organized. However, the authors should address the following criticisms:
1. Figures 1 and 2 do not include labels for peak identification.
2. The entire discussion is rather unclear. The authors often cite several plants where the different have been found, but it’s unclear why they are cited. Unless there is some similarity, or a common genus or family, this kind of information is almost useless.
3. On the other hand, previous results on seeds of this or related plants are insufficient. The authors should discuss in depth the results obtained by other researchers on guava seeds and similar plants, and how they relate with their own.
4. What kind of MS analyzer was used? Single quadrupole? Another analyzer?
5. Table 3 is not cited anywhere in the text.
6. Why were fatty acids analyzed by GC-MS, while phenolic acids and tocopherols were analyzed by HPLC-UV/DAD?
7. TPC and antioxidant activity were tested in an aqueous environment, while the extract and the oil are highly lipophilic. Thus, the tested properties could not correspond to the real activity in vivo. Why didn’t the authors test these parameters in a lipophilic environment?
8. Abbreviations should be defined at their first occurrence in the text. Here, abbreviations are defined in the Materials and Methods section, which is not the first occurrence.
9. The rats were fed GSO, not the extract. The results could be different.
10. Figure 3 is practically unreadable and thus useless.
The paper can be accepted after major revision.
Round 2
Reviewer 1 Report
I suggest to accept the manuscript
Reviewer 2 Report
The revised version of the manuscript is still unsatisfactory and I cannot recommend it for publication.
Not only am I surprised the manuscript has proceeded to next round, I am even more surprised that the authors insist on duplicate publication of a single original results on fatty acid profile of GSO. This is, as far as I understand the publication ethics, UNACCEPTABLE. Doesn’t matter whether the data presentation form differ or not. Supplementary data can eventually be published in a “data set” article but it cannot be presented as original research in the Result section of another research article. It can only be (and should be) cited in the introduction or discussed in the discussion section.
The revision is unsatisfactory, many comments implemented inaccurately, some completely ignored. Although I do not dare to judge the language, there are obviously some grammatical or typing errors and “disputable” formulations. Thus, the manuscript would greatly benefit from the language check by a native speaker. Please find my comments bellow (addressed to the Authors Response to the reviewer).
Comments to the “Response to Reviewer 2 Comments”
Response 4:
The original comment was aimed to the sentence at the line 56 of the original manuscript, not 55?:
"GSO revealed strong antioxidant, anti-low-density lipoprotein peroxidation, and anti-Gram-negative bacteria."
Response 5:
The response is confusing, stating something that is not corresponding to the revised version of the manuscript. Moreover, the authors cite Promaban et al. (2019) in the response, whereas there is different reference in the revised manuscript.
Response 8 and 5:
The ratio is usually presented as 1:x
Response 10:
The sum of total fatty acids identified should be included, not the sum of identified and unidentified FA.
Response 11:
Obviously there is still something wrong with the chromatogram. It looks if you are hiding something... I ask you kindly, if you are not capable to handle the chromatogram picture, please ask someone who is…
Figure 1:
The original comment on the figure 1 (see bellow) was not reflected.
„What were the percentage contents of the remaining peaks? What about the peak at 17.327 min, the two peaks between 29 and 31 min, the bunch of peaks between 45 and 48 min, and the group at the end of the chromatogram? Why these peaks were not identified? (I don’t mention the “invisible” part full of overlapping retention times, I consider it just integrated baseline) Some of the peaks seem to be higher than ALA and AA peaks. Please clarify.“
Response 15:
If "No", why the authentic standards are presented in the table 3? The original comment was aimed as a help in order to improve the analysis results, e.g.: identification of cpd. x was confirmed by authentic standard, other cpds. were identified tentatively. The presence of cpds. x,y,z (the authentic standards used) was not detected in the GSO..
In case the persons who carried out the MS identification are not co-authors of this manuscript it should be clearly stated where the analysis was done.
There is no mention on the GC/MS identification of phenolic compounds in the manuscript. What results are really presented then? GC/MS or HPLC/MS?
Response 16:
Where is the problem to calculate Trolox equivalents?
Response 18:
However, what we can see on the Figure 5 is unfortunately only a mixture of overlapping lines and circles...
Response 19:
The discussion is still unsatisfactory. See the original comment for details.
Response 21:
Therefore the HPLC/MS should not be included in the subsection 4.5., but new subsection on phenolic compounds identification should be created.
Response 22:
2) Then what was the reason for using corn oil for comparison? Is it supposed to have improving effect? Or the opposite?
Reviewer 3 Report
The revised version of this manuscript is still lacking under several aspects.
1. The peak labels added to Fig. 1 are apparently overlapping with peaks in the chromatogram. The results is a sub-par figure.
2. The analyte labels added to Figure 2 are too far from the corresponding peaks to be useful. I suspect several of them are also misaligned with their peaks.
3. My question still stands: If the authors analysed fatty acids and phenolic acids by GC/HPLC coupled to MS, why were tocopherols and tocotrienols analysed just by HPLC-FL? Why did fatty acids and phenolic acids need the additional selectivity and sensitivity of MS detection, and tocopherols/tocotrienols did not?
4. Measuring the antioxidant activity of an oil using a water-based assay is not correct. A lipid-based assay should be used.
5. Figure 5 is still as unreadable as Figure 3 was in the previous version. This figure should show just the means and error bars for each treatment.
6. I am not convinced that just 2 points in time are sufficient to assess possible hypocholesterolemic and/or hypotriglyceridemic effects. More points should be assessed.
7. The conclusions drawn from the TG and TC level analyses are totally unsupported by data. In particular, I am not convinced that the decrease in mean TG levels observed at 30 g LAE/kg GSO is statistically significant; even if it was, it would still not be significant in comparison to the controls, and lower than that caused by CO.
8. I am not sure that 10 rats per group are sufficient to find significant differences in TC and TG levels. The authors should have evaluated the predictive power of their test.
In my opinion, the paper still needs major revisions.